# A phase I/II study of adoptive immunotherapy using donor liver graft-derived NK cell-enriched immune cells to prevent severe infection after liver transplantation

**Masahiro Ohira**[1,2], **Yuki Imaoka**[1], **Koki Sato**[1], **Koki Imaoka**[1], **Tomoaki Bekki**[1], **Takuya Yano**[1], **Ryosuke Nakano**[1], **Hiroshi Sakai**[1], **Shintaro Kuroda**[1], **Hiroyuki Tahara**[1], **Kentaro Ide**[1], **Tsuyoshi Kobayashi**[1], **Yuka Tanaka**[1], **Junko Tanaka**[3], **Hideki Ohdan**[1]*

1 Department of Gastroenterological and Transplant Surgery, Graduate School of Biomedical and Health Sciences, Hiroshima University, Hiroshima, Japan, 2 Medical Center for Translational and Clinical Research, Hiroshima University Hospital, Hiroshima, Japan, 3 Department of Epidemiology, Infectious Disease Control and Prevention, Graduate school of Biomedical and Health Sciences, Hiroshima University, Hiroshima, Japan

* hohdan@hiroshima-u.ac.jp

**Data Availability Statement:** All relevant data are within the manuscript.

## Abstract

Bloodstream infections (BSIs) are significant postoperative complications associated with high mortality rates after liver transplantation (LT). Natural killer (NK) cells, which are key components of the innate immune system, have demonstrated potential to combat both infections and cancer. The use of activated NK cells to mitigate post-LT infections, particularly BSIs, has attracted considerable interest. We conducted a single-arm Phase I/II clinical trial to evaluate the safety and efficacy of transfusing donor liver-derived NK cells into LT recipients. Patients were administered a single infusion of these NK cells three days post-LT. The primary endpoint was BSI incidence. This study was terminated in 19 patients because of the high incidence of BSIs. Of the 19 patients receiving immunotherapy, six (31.5%) developed BSIs within one month of LT. No adverse events were directly related to NK cell infusion. Acute rejection was noted in seven patients (36.8%). After infusion, NK cell activity in the recipient's peripheral blood remained stable. In conclusion, this clinical trial did not reach the primary endpoint. This could be attributed to a significant percentage of patients presenting with high immunological risk. Nonetheless, the infusion procedure demonstrated a favorable safety profile without serious adverse events.

## Introduction

Perioperative protocols and surgical techniques for living-donor liver transplantation (LDLT) have evolved towards stabilization. However, the immediate postoperative period is also associated with infectious complications [1]. The diagnosis of such infections is often delayed or infections escalate in severity, primarily because of the necessity of immunosuppressant administration to prevent graft rejection. Bloodstream infections (BSIs), defined as bacteremia

**Funding:** This study was supported by AMED in the form of a grant awarded to HO (JP23fk0210108) and JSPS KAKENHI in the form of a grant for MO (JP23H02981). The specific roles of this author are articulated in the 'author contributions' section. The funders had no role in study design, data collection and analysis, decision to publish, or preparation of the manuscript.

**Competing interests:** NO authors have competing interests.

accompanied by clinical manifestations according to the Centers for Disease Control and Prevention (CDC) guidelines, represent the most prevalent form of infection, with local to systemic propagation. Bacterial and fungal BSIs are associated with poor prognosis, especially during the early postoperative period when patients experience profound immunosuppression. BSIs continue to contribute substantially to morbidity and mortality, with incidences ranging between 10 and 40% [2–5]. Despite the importance of post-LT prophylactic antibacterial and antifungal therapies, evidence from randomized clinical trials is scarce and a consensus on drug selection and administration methodologies remains elusive. Institutional practices vary, but a commonly adopted approach includes prophylactic administration of a 2nd or 3rd generation cephalosporin for 2–3 days [6].

Natural killer (NK) cells are integral components of innate immunity and constitute a vanguard against infectious diseases and malignancies [7]. A plethora of NK cells exhibit robust cytotoxicity against hepatoma cells, and significant cytokine production resides in the livers of both humans and mice [8, 9]. Clinical trials utilizing liver-derived NK cells to prevent hepatocellular carcinoma (HCC) recurrence have demonstrated promising safety and efficacy [10, 11]. A sub-analysis revealed that NK cell administration significantly reduces the incidence of BSI [12]. Previous studies by our group have highlighted that a single-nucleotide polymorphism in FcγRIIIA in NK cells augments post-LT BSI susceptibility because of its compromised binding properties [13]. Subsequent findings indicate the potential of NK cell therapy to offset antibody-dependent cellular cytotoxicity (ADCC) vulnerability [14].

Motivated by these findings, we initiated a clinical trial to examine an adoptive immunotherapy approach using NK cells harvested from a liver allograft perfusate to mitigate BSIs following LT.

## Materials and methods

### Study design

The clinical investigation protocol has been previously described [15]. In summary, this single-center, open-label, phase I/II trial aimed to determine the potential of living donor liver-derived NK cell immunotherapy in preventing post-LDLT BSIs. The study was conducted at Hiroshima University Hospital in Hiroshima, Japan, and participants who met the eligibility criteria were included in the study. This Phase I clinical trial was approved by the Special Committee for Regenerative Medicine, Hiroshima University, and registered with the Japan Registry of Clinical Trials (jRCTa060190036). This study was designed and conducted in accordance with the principles of the Declaration of Helsinki. Written informed consent was obtained from all patients prior to enrollment in the study. The start and end of the recruitment period of this study were between November 25, 2015, and September 6, 2022. The data were accessed for research purposes from April 1, 2023, to July 31, 2023. The authors did not have access to information that could identify individual participants during or after the data collection.

Thirty-seven patients were enrolled in the study. Qualified patients underwent standard LT and were administered enriched donor liver NK cells within three days post-LT [16]. The immunosuppression protocol comprised a 3-to 6-month regimen of tapering corticosteroids and a calcineurin inhibitor, typically tacrolimus, which was maintained with or without the addition of mycophenolate mofetil. Patients with renal insufficiency receive calcineurin inhibitor-sparing immunosuppressive regimens comprising a reduced dose of tacrolimus, corticosteroids, and mycophenolate mofetil [17]. Prophylactic antibiotics (third-generation cephalosporins) were administered perioperatively, itraconazole was administered for fungal infections, and preemptive monitoring was conducted for cytomegalovirus. The primary

endpoint of this study was the incidence of BSI within the first month post-LT. The secondary endpoints were as follows:

1. Overall survival (6 months, 1, and 3 years postoperatively)

2. Adverse events: type and severity of adverse events that occurred in this clinical study, frequency of occurrence, duration of occurrence, and causal relationship

3. Effect on immune response: Evaluation of donor-specific immunoreactivity by carboxy-fluorescein diacetate succinimidyl ester (CFSE)—mixed lymphocyte reaction (MLR) (1, 2, 3, and 4 weeks postoperatively), detection of donor-specific antibodies (annual screening), incidence of rejection, and evaluation of NK cell activity in recipient peripheral blood

4. Presence and timing of HCC recurrence (HCC cases only) and de novo carcinogenesis

5. Incidence of infectious diseases (bacterial, cytomegalovirus, fungal infections, and genetic polymorphism analysis).

## Inclusion criteria

The selection criteria for recipients (recipients of regenerative medicine) were as follows.

1. Patients undergoing partial LT for the treatment of refractory uncompensated liver cirrhosis with medical therapy

2. Patients aged $\geq$20 years at the time of consent

3. Patients who provided written consent or a substitute.

   The selection criteria for cell donors (donors) were as follows:

1. Patients who met the criteria of the LT Study Group's "Guidelines for Living Donor Surgery" and underwent living donor surgery as liver donors.

2. Patient $\geq$20 years of age at the time of obtaining consent.

3. Patients who provided written consent from the donor or surrogate to prepare liver-derived NK cells from the perfusate of the donor liver graft and administer them to the recipient.

## Exclusion criteria

The exclusion criteria were as follows:

1. Patients undergoing re-transplantation

2. Patients undergoing deceased donor LT

3. Other patients were judged by the principal investigator or sub-investigator as inappropriate for participation in this clinical research.

   The exclusion criteria for cell donors (donors) were as follows:

1. Patients judged by the principal investigator or sub-investigator were inappropriate for participation in this clinical study.

## Sample size calculation

In a previous study comparing the incidence of BSIs 1 month after LDLT in 21 patients in the NK therapy group and 21 patients in the matched non-NK therapy group for 114 living donor liver transplants performed between January 2004 and December 2009, the incidence of BSI in the NK therapy group was 10% compared to 30% in the non-NK therapy group [12]. Based on these results, the expected incidence of BSIs in NK therapy 1 month after LDLT was set at 10%, and the number of cases required to examine the threshold of 30% incidence in non-NK therapy at a 5% significance level (both sides) and power of 80% was calculated to be 34. Considering the dropout cases at the time of enrollment, we set the target number of cases to 37.

## Treatment

Cells were prepared as described previously [16]. Briefly, after organ recovery, the liver was perfused through the portal vein with 2 L of University of Wisconsin solution on a back table. Liver mononuclear cells were collected from the liver perfusate by Ficoll-Hypaque density-gradient centrifugation. Then, liver mononuclear cells were cultured with 100 U/mL human recombinant interleukin (IL)-2 (Teceleukin, Shionogi, Japan) in complete medium for three days to prime NK cells with enhanced antitumor properties. To prevent graft-versus-host disease (GVHD), anti-CD3 monoclonal antibody (mAb) (Miltenyi, Germany) was added to the culture medium (1 μg/mL) to inactivate $CD3^+$ alloreactive T cells 1 d before cell harvest. A minimum of $1 \times 10^7$ cells with a cell viability of >80% are required to release the NK cell product for infusion. After quality control verification, including Gram staining, culture, endotoxin, and mycoplasma tests, all the unfractionated cells were processed as described above and defined as the final NK cell products. NK cell products were infused intravenously into the liver transplant recipient three days after LDLT.

## Follow-up and assessment of efficacy

Blood cultures were conducted weekly for the first month post-transplantation and whenever bacteremia was suspected. Patients underwent survival checks, adverse event checks, and infection and malignancy checks for up to three years after surgery.

## Immunological assessment

All flow cytometry (FCM) analyses were performed using a BD LSR Fortessa, FACS Canto II Cytometer, and FACSCalibur flow cytometer (BD Biosciences, San Jose, CA). To detect the surface phenotype, leukocytes were stained with monoclonal antibodies: against CD3, tumor necrosis factor-related apoptosis-inducing ligand (TRAIL), CD69, NKp44, NKp46, and CD56. The data were analyzed using FlowJo software (Tree Star, Inc. Ashland, OR). A $^{51}$Cr-release assay was performed as previously described,[8] using HepG2 tumor cells (Japanese Cancer Research Resources Bank) as the targets. Briefly, $^{51}$Cr-labeled target tumor cells were added to effector cells in round-bottomed 96-well microtiter plates (BD Biosciences, San Jose, CA, USA) for 4 h at 37°C. The percentage of $^{51}$Cr released was calculated as follows: % cytotoxicity = [(cpm of experimental release–cpm of spontaneous release)/(cpm of maximum release–cpm of spontaneous release)] × 100. All assays were performed in triplicates.

To monitor immune status, an in vitro CFSE-MLR assay was performed after LT. Briefly, peripheral blood mononuclear cells prepared from the blood of the recipients, donors, and healthy volunteers with the same blood type as the donors (third-party control) for use as stimulator cells were irradiated with 30 Gy, and those obtained from the recipients for use as responder cells were labeled with 5 ml CFSE, as described previously [18]. $CD4^+$ and $CD8^+$ T-

cell proliferation and stimulation indices were quantified as previously described [18]. The diagnosis of rejection is made comprehensively by CFSE-MLR results and blood tests, including liver function, and a liver biopsy is not mandatory.

## Safety evaluation and reporting of adverse effects

After cell administration, the investigator recorded all events, regardless of whether they were related to cell administration, including complications after LT. In the event of serious complications related to prolonged hospitalization or death, the investigator promptly reported them to the Committee for Regenerative Medicine and the Ministry of Health, Labor, and Welfare.

## Statistical analysis

The incidence of BSI within the first month post-LT and its 95% confidence interval were calculated. In addition, the following statistical hypothesis test will be performed at a two-sided significance level (two-sided) of 5% by the chi-square test, using the 30% incidence of bacteremia in non-NK therapy observed in previous studies as the threshold value. A survival curve for overall survival was estimated using the Kaplan-Meier method, using the date of surgery as the starting point until the date of death (regardless of the cause of death) or the date when survival was confirmed, and the median overall survival and its 95% confidence interval were calculated. Survival rates and their 95% confidence intervals at 6 months, 1 year, and 3 years were calculated. Changes in NK cell activity at each time point were treated as paired data and compared using the Wilcoxon signed-rank test. Additionally, to compare the proportions of NK cells before and after activation, we used a paired t-test. These statistical tests were performed using the JMP software package, with a two-sided significance level set at less than 5%.

## Results

### Study population

Of the 41 potential study subjects screened for enrollment from March 2016 to April 2020, 21 were deemed eligible and enrolled. Finally, 19 LT recipients received donor liver-derived NK cell therapy (Fig 1). Reasons for exclusion included deceased donor LT (n = 13), declining to consent (n = 8), acute liver failure (n = 3), and discontinued cell production (n = 2). Table 1 shows the baseline clinical characteristics of patients. The cohort had a median age of 59 (26–70) years, and 52.6% were men. The median MELD score was 16 (7–24). ABO incompatibility and preformed donor-specific anti-HLA antibodies (DSA) were observed in 31.6% and 26.3% of the patients, respectively.

### Primary endpoint

In the examined cohort, 31.5% of the patients had bloodstream infections (BSIs) within a month after LT. This study was terminated after 19 cases due to the high incidence of BSIs. Key cases showing post-LT complications include Case 3: A patient with alcoholic cirrhosis who, pre-LT, suffered from conditions such as sepsis and pneumonia. Post-LT, a latent enterococcal infection resulted in peritonitis and bacteremia by the 12th postoperative day (Enterobacter cloacae). Case 8: A patient with NASH-induced cirrhosis who had portal vein issues before LT. On the 9th postoperative day, fever developed, and both blood and ascites cultures were positive (Enterobacter cloacae), prompting antibiotic treatment. Case 10: A patient with alcoholic and type C liver cirrhosis underwent extensive treatment for pre-LT refractory ascites. Postoperatively, they experienced thigh and pleural effusion infections, warranting antibiotic therapy. On the 12th day after LT, a blood culture test was positive (Enterococcus

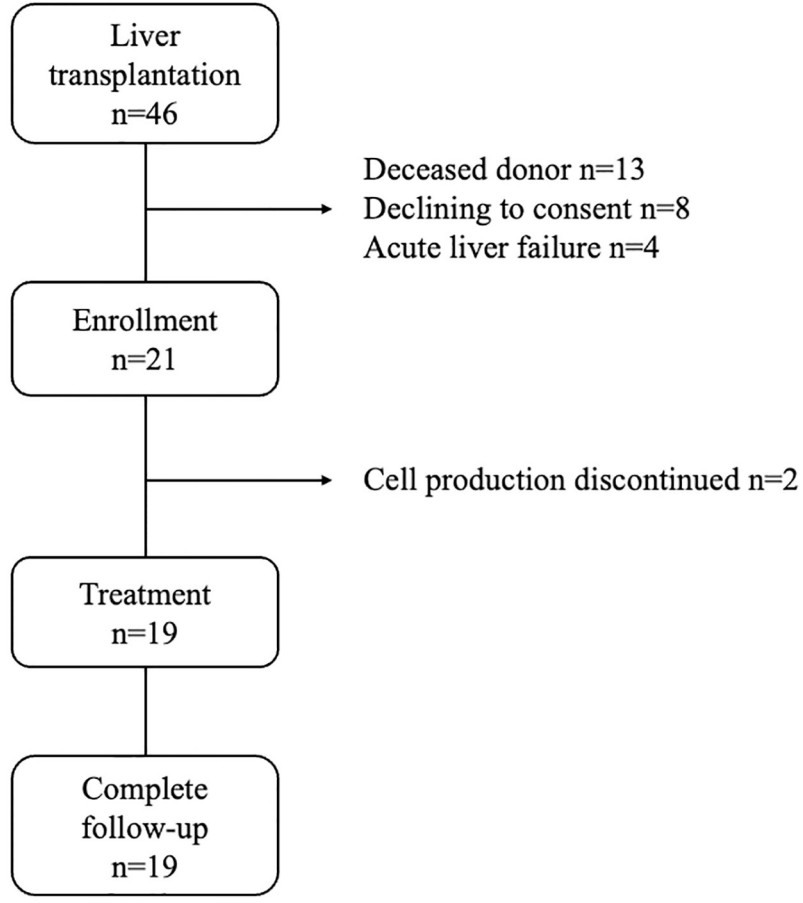

**Fig 1. Summary of trial enrolment.**

faecium), and the patient was treated with antibiotics. Case 14: An alcoholic cirrhosis patient with a history of cerebral afflictions had acute rejection post-LT. By the 15th postoperative day, the patient presented with fever and a positive blood culture result (Enterococcus faecium), necessitating antibiotics. Case 16: A patient with Budd-Chiari syndrome and cirrhosis experienced acute rejection by the 11th postoperative day. Although the patient received antibiotic treatment for the ascites infection, a subsequent fever emerged. Interestingly, their blood cultures suggested potential contamination, hinting at a false positive result (Staphylococcus aureus). Case 18: A patient with type C cirrhosis and cancer underwent LT with high-degree adhesions, leading to massive intraoperative bleeding. The postoperative period was marked by ICU admission due to severe complications (cholangitis and Enterobacter aerogenes bacteremia), and despite intensive treatment, the patient died on the 10th postoperative day.

## Secondary endpoint

The 3-year survival rate of the cohort was 73.7%. Acute graft rejection was noted in 36.8% of the patients. The patients were diagnosed with rejection based on abnormal liver function and immune monitoring using CFSE-MLR, and were treated with steroid mini-pulses, rituximab, plasma exchange, and thymoglobulin for more aggressive interventions, such as bortezomib.

**Table 1. Baseline characteristics of the patients.**

| No | Age | Sex | Etiology | MELD | ABO incompatible | Preformed DSA | Donor Age | Donor Sex | Donor relationship | HCC | Blood loss (ml) | Preoperative BSI |
|---|---|---|---|---|---|---|---|---|---|---|---|---|
| 1 | 59 | M | HCV | 14 | Yes | No | 29 | F | Child | No | 2,528 | No |
| 2 | 66 | M | HCV | 13 | Yes | No | 27 | M | Child | Yes | 4,856 | No |
| 3 | 58 | M | Alcohol | 23 | Yes | No | 55 | F | Spouse | No | 6,319 | Yes |
| 4 | 62 | F | PBC, AIH | 18 | No | Yes; B(7684), A (3169) | 32 | M | Child | No | 2,173 | No |
| 5 | 56 | F | HBV | 13 | No | No | 56 | M | Spouse | No | 4.284 | No |
| 6 | 26 | F | PSC | 16 | No | No | 55 | M | Parent | No | 2,896 | No |
| 7 | 43 | F | PBC | 24 | No | Yes; DQ(4171) | 45 | M | Sibling | No | 3,726 | No |
| 8 | 63 | F | NASH | 16 | No | Yes; A(1267) | 59 | M | Sibling | Yes | 5,125 | Yes |
| 9 | 48 | F | Alcohol, HCV | 24 | Yes | No | 24 | F | Child | No | 5,000 | No |
| 10 | 65 | M | HCV | 20 | Yes | No | 38 | M | Child | No | 5,965 | Yes |
| 11 | 62 | M | Alcohol | 18 | No | No | 57 | F | Sibling | Yes | 2,950 | No |
| 12 | 58 | M | Alcohol | 15 | No | No | 31 | M | Child | No | 2,740 | No |
| 13 | 67 | F | HCV | 15 | No | No | 41 | M | Child | Yes | 3,988 | No |
| 14 | 46 | F | Alcohol | 13 | Yes | No | 45 | M | Spouse | No | 1,806 | Yes |
| 15 | 46 | M | Alcohol | 20 | No | No | 44 | F | Spouse | No | 2,030 | No |
| 16 | 33 | M | Budd-Chiary syndrome | 10 | No | No | 64 | F | Parent | No | 1,733 | Yes |
| 17 | 69 | M | Budd-Chiary syndrome | 17 | No | No | 69 | M | Child | No | 2,417 | No |
| 18 | 66 | M | HCV | 7 | No | No | 66 | M | Child | Yes | 20,218 | Yes |
| 19 | 70 | F | PBCl | 14 | No | No | 45 | M | Child | Yes | 1,216 | No |

Abbreviations; AIH, autoimmune hepatitis; DSA, donor specific antibody; HBV, hepatitis B; HCC, hepatocellular carcinoma; HCV, hepatitis C; MELD, The Model of End-stage Liver Disease; NASH, nonalcoholic steato-hepatitis; PBC, Primary biliary cholangitis; PSC, Primary Sclerosing Cholangitis N/A; not applicable,

A liver biopsy was performed in only one case, and the diagnosis was moderate acute rejection (Case 1). Case 14 experienced a severe reaction to thymoglobulin, resulting in cardiodynamic instability and subsequent death due to cerebral hemorrhage. Case 19: Despite the low preoperative anti-donor antibody levels, the patient experienced acute rejection post-LT. Interventions such as steroid pulse therapy, plasma exchange, and thymoglobulin administration failed to stabilize the patient, leading to death.

Of the 19 patients, six had hepatocellular carcinoma (HCC) but adhered to the Milan criteria. Currently, there has been no recurrence of HCC, with a median observation period of 54.2 months. De novo carcinogenesis manifested in one patient as early stage gastric cancer 54.4 months post-LT. Additionally, 84.2% of the patients had bacterial infections, 52.6% had cytomegalovirus infections, and 42.1% had fungal infections.

## Characteristics of liver NK cells

Liver perfusion produced $665 \times 10^6$ cells. After culturing, the recovery rate was 59.6%, leading to administration of $364 \times 10^6$ cells. The cells displayed a high viability of 92.9% (Table 2). Before culture, NK cells constituted 23.9% of the total cells, which increased to 33.2% after culture. The activation markers were CD69 (92.2%), TRAIL (37.3%), NKp44 (22.4%), NKp46 (75.7%), and cytotoxic activity (109.0%) (Fig 2).

**Table 2. Characteristics of final product.**

| No | Graft volume (g) | Cell Dose (×10$^8$) | Viability (%) | Cytotoxicity (%) | Purity NK cell (%) | TRAIL on NK cell (%) | Purity T cell (%) | Endotoxin | Mycoplasma | Culture test |
|---|---|---|---|---|---|---|---|---|---|---|
| 1 | 574 | 872 | 94.5 | 101.3 | 54.9 | 61.4 | 0.00 | negative | negative | negative |
| 2 | 450 | 662 | 84.9 | 94.7 | 49.3 | 78.9 | 0.01 | negative | negative | negative |
| 3 | 642 | 421 | 89.7 | 65.0 | 16.2 | 20.3 | 0.03 | negative | negative | negative |
| 4 | 478 | 673 | 86.4 | 103.9 | 22.7 | 26.3 | 0.05 | negative | negative | negative |
| 5 | 608 | 700 | 81.9 | 100.7 | 40.4 | 32.4 | 0.00 | negative | negative | negative |
| 6 | 474 | 131 | 88.0 | 186.6 | 22.6 | 38.9 | 0.05 | negative | negative | negative |
| 7 | 502 | 407 | 98.1 | 125.6 | 36.0 | 42.0 | 0.02 | negative | negative | negative |
| 8 | 462 | 240 | 95.9 | 112.5 | 37.7 | 31.6 | 0.02 | negative | negative | negative |
| 9 | 492 | 330 | 91.7 | 156.6 | 39.2 | 46.5 | 0.02 | negative | negative | negative |
| 10 | 564 | 582 | 95.3 | 84.6 | 32.4 | 37.3 | 0.01 | negative | negative | negative |
| 11 | 636 | 394 | 97.0 | 102.0 | 52.1 | 23.5 | 0.01 | negative | negative | negative |
| 12 | 406 | 319 | 91.3 | 86.8 | 37.1 | 42.3 | 0.00 | negative | negative | negative |
| 13 | 462 | 278 | 97.3 | 129.9 | 33.3 | 35.7 | 0.01 | negative | negative | negative |
| 14 | 534 | 295 | 98.7 | 53.1 | 35.7 | 15.3 | 0.01 | negative | negative | negative |
| 15 | 616 | 265 | 92.6 | 179.9 | 22.3 | 55.8 | 0.04 | negative | negative | negative |
| 16 | 450 | 117 | 93.5 | 98.2 | 26.7 | 32.9 | 0.05 | negative | negative | negative |
| 17 | 338 | 84 | 96.7 | 104.3 | 23.7 | 44.2 | 0.25 | negative | negative | negative |
| 18 | 560 | 257 | 95.4 | 74.1 | 27.5 | 22.9 | 0.07 | negative | negative | negative |
| 19 | 362 | 118 | 95.5 | 113.6 | 25.4 | 31.3 | 0.09 | negative | negative | negative |

Abbreviations: NK cell; Natural killer cell, TRAIL; TNF-related apoptosis inducing ligand

## Safety and toxicity profile

There are no reported ailments associated with treatment. Adverse events after LT, as categorized by NCI-CTC-AE, included 23 Grade I elements (53 instances), 62 Grade II elements (202 occurrences), 48 Grade III elements (161 instances), and nine Grade IV elements (17 instances), as detailed in Table 3. All adverse events including infections, rejection, and complications related to LT were thoroughly reviewed by the Efficacy and Safety Evaluation Committee. This committee evaluated whether each event was likely to be associated with NK cell infusion or LT. Based on these evaluations, no adverse events were directly attributed to NK cell therapy, as all were complications typical of transplantation.

## Immunological assessments

The immune status of the recipient was evaluated using CFSE-MLR. Seven patients were suspected of having rejection in the MLR category [19], and their liver function had deteriorated; therefore, we diagnosed them as having a rejection reaction and administered steroid pulse therapy. Pre-existing DSAs were identified in 21.1% of the patients. All cases that tested positive for Class I disappeared, but a Class II-positive case remained for as long as three years post-LT. After the operation, 21.1% (or four patients) manifested fresh de novo DSAs; one turned from positive to negative for de novo DSA Class I, while three Class II cases continued. Every patient presented with regular liver function and no instances of antibody rejection. After NK cell infusion, there was an increase in CD69 and TRAIL expression in peripheral blood NK cells, reaching a height at two weeks. The cytotoxic response of the peripheral blood lymphocytes remained largely constant (Fig 3). FcγRIIIA SNPs in the recipient and donor

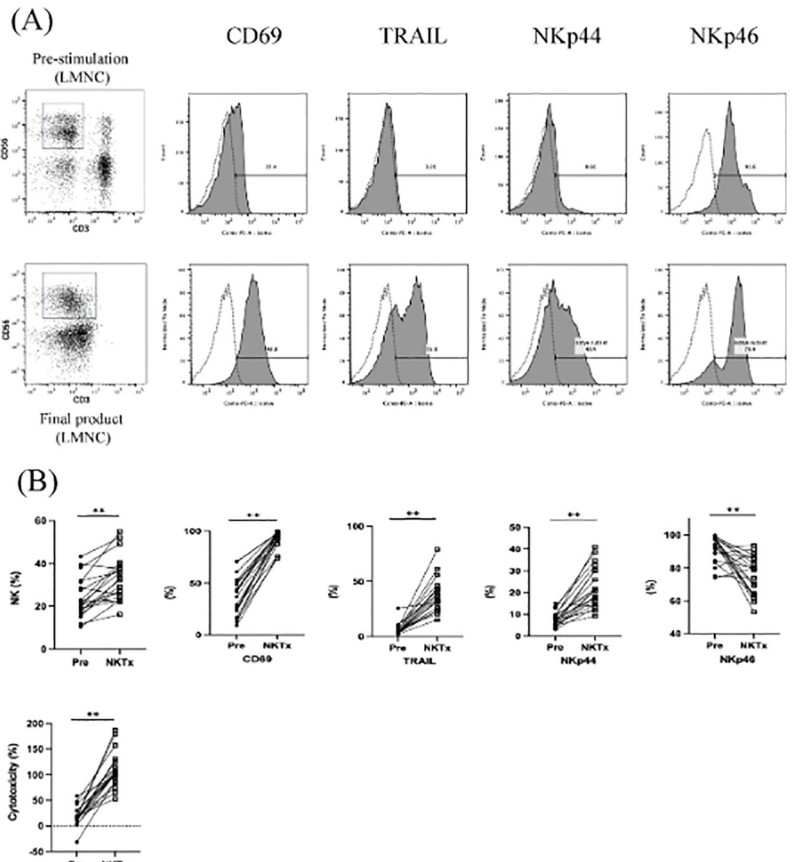

**Fig 2. Characteristics of liver NK cells.** (A) Representative NK cell phenotypes. The upper panel displays pre-stimulation, whereas the lower panel represents the final product. Histograms show activation markers specifically gated on CD3-CD56+ NK cells. Transparent graphs correspond to isotype controls. (B) The proportions of NK cells before and after activation, expression intensity of activation markers, and cytotoxic activity for each case. To compare the proportions of NK cells before and after activation, we used a paired t-test. Asterisk (*) denotes a statistically significant difference (p < 0.01).

groups are shown in Table 4. Five of the eight cases (62.5%) in which both the recipient and donor were F carriers developed BSIs.

## Discussion

The primary aim of this phase I/II trial was to assess the safety and potential efficacy of donor-derived NK cell therapy in reducing post-LT BSIs. Safety was the focus and efficacy was evaluated against historical control data. Adverse events were monitored, categorized by severity, and reviewed by the Efficacy and Safety Evaluation Committee to determine their relationship with NK cell therapy. Although the study lacked a control group, we compared the incidence of BSI with historical data from our institution and other published studies. The trial aimed to enroll 37 patients, but ended prematurely in 19 patients owing to failure to achieve the primary endpoint. No adverse events related to cell administration were detected, indicating the achievement of a safety-oriented secondary endpoint.

Numerous post-LT BSI risk factors exist, and a meta-analysis revealed 11 significant factors, including male sex, ascites, MELD score, Child-Pugh class C, operation time, incompatible

**Table 3. Summary of adverse events after liver transplantation.**

| | Grade I | Grade II | Grade III | Grade IV |
|---|---|---|---|---|
| Hyponatremia | 8 (42) | 4 (21) | | |
| Thrombocytopenia | 5 (26) | 1 (5) | 11 (58) | |
| Hyperbilirubinemia | 5 (26) | 6 (32) | 6 (32) | |
| Renal dysfunction | 4 (21) | 8 (42) | | |
| Hypocalcemia | 4 (21) | 2 (11) | 1 (5) | |
| Hyperkalemia | 3 (16) | 6 (32) | | |
| Liver dysfunction | 3 (16) | 13 (68) | 3 (16) | |
| Hyperammonemia | 3 (16) | 8 (42) | | |
| Pancytopenia | 2 (11) | 2 (11) | 1 (5) | |
| Hypermagnesemia | 2 (11) | | | |
| Hypokalemia | 2 (11) | 4 (21) | 1 (5) | |
| Coagulation factor abnormalities | 1 (5) | | 6 (32) | |
| Low serum PT | 1 (5) | 2 (11) | | |
| Anemia | 1 (5) | 9 (47) | 15 (79) | |
| Pleural fluid | 1 (5) | 3 (16) | 10 (53) | |
| Metabolic acidosis | 1 (5) | 2 (11) | 2 (11) | |
| Atelectasis | 1 (5) | | | |
| Portal vein thrombus | 1 (5) | | 1 (5) | |
| Hoarseness | 1 (5) | | | |
| Hypoalbuminemia | 1 (5) | 15 (79) | 1 (5) | |
| Pancreatitis | 1 (5) | 1 (5) | 1 (85) | |
| Lymphocytopenia | 1 (5) | | | |
| Malnutrition | 1 (5) | | | |
| Low serum AT3 | | 12 (63) | 10 (53) | |
| Cytomegalovirus infection | | 8 (42) | 2 (11) | |
| Paralytic ileus | | 7 (37) | | |
| Biliary enzyme elevation | | 6 (32) | 1 (5) | |
| Constipation | | 6 (32) | | |
| Hyperglycemia | | 6 (32) | | |
| Hypothyroidism | | 5 (26) | | |
| Hypertension | | 5 (26) | | |
| Ascites | | 5 (26) | 8 (42) | |
| Hyperuricemia | | 4 (21) | | |
| Hypoglobulinemia | | 3 (16) | 9 (47) | |
| Zinc Deficiency | | 2 (11) | | |
| Hypomagnesemia | | 2 (11) | 1 (5)c | |
| Osteoporosis | | 2 (11) | | |
| Diarrhea | | 2 (11) | | |
| Fungal infections | | 2 (11) | | |
| Abdominal cavity infection | | 2 (11) | 1 (5) | |
| Blood pressure decrease | | 2 (11) | 3 (16) | |
| Cholangitis | | 2 (11) | 3 (16) | |
| Spontaneous Bacterial Peritonitis | | 2 (11) | 6 (32) | |
| Liver congestion | | 2 (11) | | |
| Intravascular dehydration | | 1 (5) | | |
| Bleeding | | 1 (5) | 1 (5) | |
| Iron Deficiency Disorders | | 1 (5) | | |

*(Continued)*

**Table 3.** (Continued)

| | Grade I | Grade II | Grade III | Grade IV |
|---|---|---|---|---|
| Dyslipidemia | | 1 (5) | | |
| Hypofolatemia | | 1 (5) | | |
| Neutropenia | | 1 (5) | 1 (5)bilia | |
| Albuminuria | | 1 (5) | | |
| Influenza infection | | 1 (5) | | |
| Gout | | 1 (5) | | |
| Oliguria | | 1 (5) | | |
| Pulmonary edema | | 1 (5) | 4 (21) | 1 (5) |
| Bone fracture | | 1 (5) | 1 (5) | |
| Postoperative pain | | 1 (5) | | |
| Functional dyspepsia | | 1 (5) | | |
| Subcutaneous hematoma | | 1 (5) | | |
| Aspergillus pneumonia | | 1 (5) | | |
| Insomnia | | 1 (5) | | |
| HCV infection | | 1 (5) | | |
| Biliary fistula | | 1 (5) | | |
| Cerebrovascular disease | | 1 (5) | | |
| Hypoglycemia | | 1 (5) | | |
| Urinary infection | | 1 (5) | 8 (42) | |
| Blood stream infection | | 1 (5) | 5 (26) | |
| Liver infarction | | 1 (5) | | |
| Acute rejection | | | 8 (42) | |
| Pneumonia | | | 4 (21) | |
| Gastrointestinal bleeding | | | 4 (21) | |
| Acute circulatory failure | | | 2 (11) | 1 (5) |
| Respiratory failure | | | 2 (11) | 2 (11) |
| Disseminated intravascular coagulation | | | 2 (11) | 1 (5) |
| Acute renal failure | | | 2 (11) | 5 (26) |
| Thrombotic microangiopathy | | | 2 (11) | |
| Vomiting | | | 1(5) | |
| Gastroenteritis | | | 1(5) | |
| Delirium | | | 1(5) | |
| Suicide attempt | | | 1(5) | |
| Pneumothorax | | | 1(5) | |
| Hypercalcemia | | | 1(5) | |
| Wound infection | | | 1(5) | |
| Pneumocystis carinii pneumonia | | | 1(5) | 1 (5) |
| Drug-induced psychotic disorder | | | 1(5) | |
| Septic shock | | | | 3 (16) |
| Liver failure | | | | 2 (11) |
| Hemorrhagic shock | | | | 1 (5) |

The data are presented as No. (%). The data included Adverse Events reported after transplantation. None of the events was related to the decision of the Data Safety Monitoring Board.

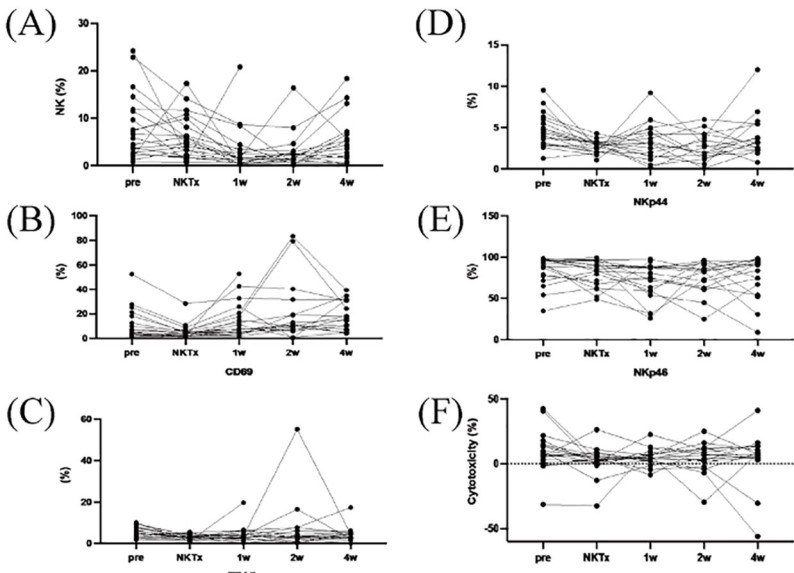

**Fig 3. Changes in peripheral blood NK cell counts in recipients.** (A) Percentage of NK cells, (B) CD69, (C) TRAIL, (D) NKp44, and (E) NKp46 expression levels, and (F) cytotoxic activity in recipient peripheral blood for each case. Changes in NK cell activity at each time point were treated as paired data and compared using the Wilcoxon signed-rank test.

blood type, operative blood loss, rejection, biliary complications, hemodialysis, and retransplantation [20]. However, no association between these factors and the BSI was established in this study. Previously, we have highlighted the role of recipient FcγRIIIA SNPs in bacteremia and the effectiveness of NK cell therapy against vulnerable SNPs [13, 14]. Notably, all six BSI-affected patients in this study possessed donor FcγRIIIA SNP of the F carrier type (Table 4), suggesting that the administered NK cells may have been susceptible to infection. Our study revealed that both recipient and donor genetic profiles, particularly FcγRIIIA F-carrier status, play a crucial role in the risk of post-LT BSIs. Recipients with this status are more susceptible to BSIs, and when donor NK cells carry the F allele, the effectiveness of NK cell therapy is reduced. We propose several steps to address this: stratifying patients by immunological risk, focusing on low-risk recipients without predisposing factors such as FcγRIIIA F-carrier status, and increasing infection monitoring for genetically susceptible patients. Adjusting prophylactic treatments such as broad-spectrum antimicrobials may be necessary for high-risk individuals. We also plan to conduct a new randomized controlled clinical trial (RCT) to evaluate NK cell therapy while exploring the genetics and functionality of donor NK cells, their interaction with the recipient's immune system, and strategies to improve NK cell homing and persistence in the liver to prevent infection.

Post-LT acute rejection (AR) is associated with numerous risk factors including DSA-positive pre-LT [21], infection [22], and pre-existing autoimmune disorders [23]. This study

**Table 4. Association of FcγRIIIA single nucleotide polymorphisms with blood stream infection.**

| Recipient | F carrier | | VV | |
|---|---|---|---|---|
| Donor | F carrier | VV | F carrier | VV |
| Blood stream infection | 5/8 | 0/5 | 1/1 | 0/5 |

recorded a high AR prevalence, with seven of 19 patients (36.8%), likely attributable to the risk factors observed in cases of ABO-incompatible transplants, autoimmune disease, and heightened preoperative anti-donor response. CFSE-MLR used to diagnose rejection, and rejection treatment is performed without liver biopsy. Therefore, the incidence of rejection may be higher than that reported in the previous studies. Adverse events were evaluated by the Efficacy and Safety Evaluation Committee, who concluded that the treatment did not increase the incidence of rejection compared to previous cases. Many patients had high immunological risk factors, which likely contributed to these events rather than to the therapy itself. However, it is difficult to scientifically prove that administered cells are unrelated to adverse events, including rejection. Future RCTs should be conducted to confirm this hypothesis.

We retrieved $665 \times 10^6$ liver-derived NK cells from the donor grafts and administered a median of $364 \times 10^6$ cells. The expression of activation markers, including TRAIL, and cytotoxic activity against hepatic carcinoma cell lines were satisfactory. This aligns with the results of our previous phase I trial involving LT for HCC [10, 16]. Although the incidence of LT in HCC is decreasing, recurrence remains a significant challenge. In the future, it will be necessary to develop cell therapies that target the inhibition of HCC recurrence.

This study has several limitations. Indeed, the cell product was not purely composed of NK cells as it contained other immune cells such as T cells, B cells, macrophages, and dendritic cells. However, anti-CD3 monoclonal antibodies were used to remove alloreactive T cells prior to administration to mitigate the risk of GVHD. The focus of this therapy is on the potential of NK cells to prevent infection; however, we acknowledge the presence of other immune cells in the product. Next, without a control group, definitive conclusions regarding the effects of NK cell therapy on peripheral NK cells are limited. However, we monitored NK cell activity in peripheral blood and observed that, while there was no significant enhancement in cytotoxic activity, there was an increase in activation markers such as CD69 and TRAIL two weeks after the infusion. These data suggest that the therapy did not significantly alter the cytotoxic function of peripheral NK cells. We have previously reported that liver-resident NK cells home to the liver via chemokines and chemokine receptors [24]. Therefore, we hypothesized that the infused NK cells likely migrated to the liver and exerted their effects locally, rather than systemically. Finally, although this trial was designed as a single-arm phase 1/2 trial, there was a bias compared to previous cases because many immunologically high-risk patients and patients with poor preoperative conditions were enrolled. To confirm the therapeutic effects and adverse events of this cell therapy, it will be necessary to plan an RCT in the future.

In conclusion, while NK cell administration was deemed safe, it failed to prevent postoperative bacteremia in patients with high immunologic risk undergoing living donor LT. This could be attributed to the high proportion of high-risk patients, AR occurrence, and the specific therapeutic regimen applied.

## Supporting information

**S1 File.**
(PDF)

**S2 File.**
(PDF)

**S1 Checklist. SPIRIT 2013 checklist: Recommended items to address in a clinical trial protocol and related documents\*.**
(DOC)

**S2 Checklist.**
(XLSX)

## Acknowledgments

We would like to thank Editage for proofreading this manuscript.

## Author Contributions

**Conceptualization:** Masahiro Ohira, Hideki Ohdan.

**Data curation:** Masahiro Ohira, Yuki Imaoka, Koki Sato, Koki Imaoka, Tomoaki Bekki, Takuya Yano, Ryosuke Nakano, Hiroyuki Tahara.

**Formal analysis:** Junko Tanaka.

**Investigation:** Masahiro Ohira, Yuki Imaoka, Koki Sato, Koki Imaoka, Tomoaki Bekki, Takuya Yano, Ryosuke Nakano, Hiroshi Sakai, Shintaro Kuroda, Kentaro Ide, Tsuyoshi Kobayashi, Yuka Tanaka.

**Methodology:** Yuka Tanaka.

**Supervision:** Hideki Ohdan.

**Writing – original draft:** Masahiro Ohira, Yuki Imaoka.

**Writing – review & editing:** Masahiro Ohira, Hideki Ohdan.

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
