## [Decision Letter · Decision Letter 0]

12 Aug 2024

PONE-D-24-28239A phase I/II study of adoptive immunotherapy using donor liver graft-derived natural killer cells to prevent severe infection after liver transplantationPLOS ONE

Dear Dr. Ohira,

Thank you for submitting your manuscript to PLOS ONE. After careful consideration, we feel that it has merit but does not fully meet PLOS ONE’s publication criteria as it currently stands. Therefore, we invite you to submit a revised version of the manuscript that addresses the points raised during the review process.

We look forward to receiving your revised manuscript.

Kind regards,

Pavel Strnad

Academic Editor

PLOS ONE

Journal Requirements:

6. Please include a caption for figure 3. 

7. Please include your tables as part of your main manuscript and remove the individual files. Please note that supplementary tables (should remain/ be uploaded) as separate "supporting information" files

Reviewers' comments:

Reviewer's Responses to Questions

**Comments to the Author**

1. Is the manuscript technically sound, and do the data support the conclusions?

Reviewer #1: Yes

Reviewer #2: Yes

2. Has the statistical analysis been performed appropriately and rigorously? 

Reviewer #1: Yes

Reviewer #2: Yes

3. Have the authors made all data underlying the findings in their manuscript fully available?

Reviewer #1: Yes

Reviewer #2: No

4. Is the manuscript presented in an intelligible fashion and written in standard English?

Reviewer #1: Yes

Reviewer #2: Yes

5. Review Comments to the Author

Reviewer #1: Thank you for letting me review the manuscript "A phase I/II study of adoptive immunotherapy using donor liver graft-derived natural killer cells to prevent severe infection after liver transplantation" by Ohira et al.

The authors attempted to evaluate the hypothesis that a transfer of donor liver derived NK cells could reduce this risk of blood stream infections (BSI) after liver transplantation. Unfortunately, the analyses displayed a higher rate of BSI than expected during study conception. The autors attribute this to the high immunological risk of the recipients, as indicated by a high rate of AB0 incompatbile transplants and of acute rejections after transplantation.

The study design is intriguing, especially as there are also hopes for cellular therapy refining immunosuppressive regimens after transplantation. The high rate of BSIs is indeed surprising, as is the high rate of acute rejection.

I have a few questions, that the authors should/could comment on:

1. With regard to the high rate of acute rejections - why was mycophenolate mofetil not applied in all recipients?

2. Did you perform liver biopsies per-protocol to determine rejections or as a result of elevated liver enzymes, bilirubin etc. ?

3. What kind and degree of acute rejection did you observe in your recipients? If you performed per-protocol biopsies were these clinically relevant?

4. How can you be sure that the adoptive transfer of allogeneic cells is not related to the higher incidence of acute rejections as adverse effect?

5. If the high rate of BSI is related to the high immunological risk, did you apply more immunosuppressants in these recipients, compared to your low immunological risk patients?

6. What is your prophylactic approach for bacterial, fungal and viral infections?

7. Which days postoperatively did you draw blood cultures to determine BSI?

8. What bacteria did you find in your patients with BSI?

9. Where all BSI clinically relevant?

10. Did you perform donor bile swabs during donor partial hepatectomy or recipient bile swabs during recipient hepatectomy to gather information on the microbial spectrum of the donor/recipient?

11. Did you change your prophylactic approach for infections or your immunosuppressive regimen with regard to the results of your study?

12. Could you elaborate on what steps are required to further evaluate donor NK cell therapy in improving posttransplant infections? Are you planning on performing a new study in immunological low risk recipients?

A minor remark:

I would remove exclusion criterion number 1 for the cell donors, as it is already mentioned under general exclusion criteria.

Reviewer #2: Ohira and co-authors present a interesting manuscript on a phase I/II study of cell-based immunotherapy to prevent bacteraemic infections after liver transplant.

While the concept is innovative and the study itself seems to be well done, there are a number of issues that need to be clarified:

- what was the aim of this study? What kind of signal would the authors consider as an adverse event associated with the immunotherapy? How did they authors aim to assess the endpoints without any controll-group?

- concerning safety: how did the authors judge that all the adverse events were not related to the type of therapy?

- concerning the therapy: the authors need to specify much more clearly how the cell product was produced and when it was applied. I appreciate that the authors provide this information in a previous publication and in the supplementary material, but this issue is so fundamental to the whole manuscript that it should be included in the main part

- the authors describe their therapy mainly as NK cell therapy. However, it seems that their product was rather modestly enriched for NK cells, but included a substantial number of non-NK cells

- what are the characteristics of the peripheral NK cells to show if their is no controll group? Their does not seem to be much change after infuscion of the cells, which might be expected. Of course, intrahepatic NK cells differ substantially from peripheral NK cells.

- The discussion is poor. The authors should write a proper discussion, including technical aspects (e.g. would repeated application be better, was the timing corret etc.), and aspects of future approaches (do the authors think that their approach does not work at all, why did they not achieve their goals, what improvements can be made or is the concept by itself flawed? Can "better" cell products be achieved? Where do the cells home to?)

- a lot of abbreviations are not explained in the text; it is unclear what "Sol-Medrol" is and so on.

- the english of the section "immunological assessments" is not very clear; I do not understand all sentences

6. PLOS authors have the option to publish the peer review history of their article (what does this mean?). If published, this will include your full peer review and any attached files.

Reviewer #1: No

Reviewer #2: No

---

## [Author Response · Author response to Decision Letter 0]

18 Oct 2024

Dear Professor, Strnad:

Thank you for providing the reviewers’ insightful comments on our manuscript, titled “A phase I/II study of adoptive immunotherapy using donor liver graft-derived natural killer cells to prevent severe infection after liver transplantation.” We appreciate the opportunity to resubmit our revised manuscript for reconsideration for publication in PLOS ONE. Below is a detailed point-by-point response to the reviewers' queries. All modifications are marked in red in the revised manuscript.

Response to Reviewer #1:

Query 1: With regard to the high rate of acute rejections - why was mycophenolate mofetil not applied in all recipients?

Reply:

The standard protocol involves the use of steroids and Prograf. In cases of impaired renal function, we administer a combination of steroids, Prograf, and MMF to reduce Prograf concentrations. We have added a note regarding this protocol to the Methods section (P.5 lines 13-15).

Query 2: Did you perform liver biopsies per-protocol to determine rejections or as a result of elevated liver enzymes, bilirubin etc. ?

Reply:

At our facility, we use CFSE-MLR for immune monitoring alongside liver function assessment. Liver biopsies were conducted in only two of the seven cases, and one case displayed moderate rejection. A description of CFSE-MLR has been included in the Methods section (P.10 lines 10-15).

Query 3: What kind and degree of acute rejection did you observe in your recipients? If you performed per-protocol biopsies were these clinically relevant?

Reply:

Only one biopsy-confirmed case of rejection occurred, but four out of seven cases exhibited steroid-resistant rejection, requiring thymoglobulin treatment.

We added the following sentences in the result section (Page 15 lines 5-9).

The patients were diagnosed with rejection based on abnormal liver function and immune monitoring using CFSE-MLR and were treated with steroid mini-pulses, rituximab, plasma exchange, and thymoglobulin to more aggressive interventions, such as bortezomib. A liver biopsy was performed in only one case, and the diagnosis was moderate acute rejection (Case 1).

Query 4: How can you be sure that the adoptive transfer of allogeneic cells is not related to the higher incidence of acute rejections as adverse effect?

Reply:

Adverse events were evaluated by the Efficacy and Safety Evaluation Committee, which concluded that the treatment did not increase the incidence of rejection compared to previous cases. Many patients had high immunological risk factors, which likely contributed to these events rather than the therapy itself.

These sentences were added to the discussion section (Page 24 lines 1-4).

Query 5: If the high rate of BSI is related to the high immunological risk, did you apply more immunosuppressants in these recipients, compared to your low immunological risk patients?

Reply:

Immunosuppressive regimens were not adjusted based on immunological risk in this study.

Query 6: What is your prophylactic approach for bacterial, fungal and viral infections?

Reply:

Prophylactic antibiotics (third-generation cephalosporins) were administered during the perioperative period, itraconazole was given for fungal infections, and preemptive monitoring was conducted for cytomegalovirus. These details have been added to the Methods section (P.5 lines 15-17).

Query 7: Which days postoperatively did you draw blood cultures to determine BSI?

Reply:

Blood cultures were conducted weekly for the first month post-transplantation and whenever bacteremia was suspected. This sentence was added in the Method section (P.9 line 13-14).

Query 8: What bacteria did you find in your patients with BSI?

Reply:

Six cases of bacteremia were observed, with two cases each of Enterobacter cloacae and Enterococcus faecium, and one case each of Staphylococcus aureus and Enterobacter aerogenes. These results have been added to the Results section (P.13-15).

Query 9: Where all BSI clinically relevant?

Reply:

Two cases resulted in septic shock, three improved with antibiotics, and one was suspected contamination.

Query 10: Did you perform donor bile swabs during donor partial hepatectomy or recipient bile swabs during recipient hepatectomy to gather information on the microbial spectrum of the donor/recipient?

Reply:

Bile cultures were not performed, but this approach will be considered in future studies.

Query 11: Did you change your prophylactic approach for infections or your immunosuppressive regimen with regard to the results of your study?

Reply:

We agree that our infection prevention and immunosuppressant protocols need revision based on the results of this study.

Query 12: Could you elaborate on what steps are required to further evaluate donor NK cell therapy in improving posttransplant infections? Are you planning on performing a new study in immunological low risk recipients?

Reply:

Thank you for this important question. Our study revealed that both recipient and donor genetic profiles, particularly FcγRIIIA F-carrier status, play a crucial role in the risk of Post-LT BSIs. Recipients with this status are more susceptible to BSIs, and when donor NK cells also carry the F allele, the effectiveness of NK cell therapy is reduced.

We propose several steps to address this: stratifying patients by immunological risk, focusing on low-risk recipients without predisposing factors, such as FcγRIIIA F-carrier status, and increasing infection monitoring for genetically susceptible patients. Adjusting prophylactic treatments, such as broader-spectrum antimicrobials, may be necessary for high-risk individuals. We also plan to conduct a new clinical trial in a low-risk population to evaluate NK cell therapy while exploring the genetics and functionality of donor NK cells, their interaction with the recipient’s immune system, and strategies to improve NK cell homing and persistence in the liver to enhance infection prevention.

These sentences were added in the discussion section (P.23 lines 4-14).

Query 13: I would remove exclusion criterion number 1 for the cell donors, as it is already mentioned under general exclusion criteria.

Reply:

Thank you for pointing this out. We have removed the redundant exclusion criterion.

Response to Reviewer #2

Query 1: What was the aim of this study? What kind of signal would the authors consider as an adverse event associated with the immunotherapy? How did they authors aim to assess the endpoints without any controll-group?

Reply:

The primary aim of this phase I/II trial was to assess the safety and potential efficacy of donor-derived NK cell therapy for reducing bloodstream infections (BSI) after liver transplantation. Safety was the focus, and efficacy was evaluated against historical control data. Adverse events were monitored, categorized by severity, and reviewed by the Efficacy and Safety Evaluation Committee to determine any relationship to the NK cell therapy. Although the study lacked a control group, we compared the incidence of BSI with historical data from our institution and other published studies. This has been clarified in the Discussion section(P.22 lines 3-8).

Query 2: Concerning safety: how did the authors judge that all the adverse events were not related to the type of therapy?

Reply:

All adverse events, including infections, rejections, and complications related to liver transplantation, were thoroughly reviewed by the Efficacy and Safety Evaluation Committee. This committee evaluated whether each event was likely to be associated with the NK cell infusion or with the liver transplantation itself. Based on these evaluations, no adverse events were directly attributed to the NK cell therapy, as all were considered typical complications of the transplantation process. We have added a more detailed explanation of this evaluation process in the revised manuscript under the Safety section (P.17 lines 9- P.18 lines 5).

Query 3: Concerning the therapy: the authors need to specify much more clearly how the cell product was produced and when it was applied. I appreciate that the authors provide this information in a previous publication and in the supplementary material, but this issue is so fundamental to the whole manuscript that it should be included in the main part

Reply:

We agree that providing more detail on the cell product is essential for clarity. In the revised manuscript, we have expanded the Methods section to describe the preparation of the NK cells in greater detail. The liver mononuclear cells were harvested from the liver perfusate using Ficoll-Hypaque density-gradient centrifugation. They were then cultured with human recombinant IL-2 for three days to activate NK cells. Anti-CD3 monoclonal antibodies were added one day prior to harvest to deplete T cells, which could cause graft-versus-host disease (GVHD). A minimum of 1 × 10^7 cells with a viability of >80% were required for infusion. After passing quality control tests (e.g., Gram staining, endotoxin, and mycoplasma tests), the final product was infused into recipients three days post-transplantation. These details are now clearly described in the manuscript. (Page 8 lines 16- Page 9 lines 11)

Query 4: The authors describe their therapy mainly as NK cell therapy. However, it seems that their product was rather modestly enriched for NK cells, but included a substantial number of non-NK cells

Reply:

Thank you for this important observation. Indeed, the cell product was not purely composed of NK cells as it contained other immune cells such as T cells, B cells, macrophages, and dendritic cells. However, to mitigate the risk of GVHD, anti-CD3 monoclonal antibodies were used to remove alloreactive T cells prior to administration. The focus of this therapy was on the potential of NK cells to prevent infection; however, we acknowledge the presence of other immune cells in the product. This clarification has been added to the revised manuscript(P.24 lines 11-16).

Query 5: What are the characteristics of the peripheral NK cells to show if their is no controll group? Their does not seem to be much change after infuscion of the cells, which might be expected. Of course, intrahepatic NK cells differ substantially from peripheral NK cells.

Reply:

Without a control group, definitive conclusions regarding the effect of NK cell therapy on peripheral NK cells are limited. However, we monitored NK cell activity in peripheral blood and observed that, while there was no significant enhancement in cytotoxic activity, there was an increase in activation markers such as CD69 and TRAIL two weeks after the infusion. These data suggested that the therapy did not significantly alter the cytotoxic function of peripheral NK cells, although we hypothesized that the infused NK cells likely migrated to the liver and exerted their effects locally rather than systemically. We have further elaborated on this hypothesis in the Discussion section of the revised manuscript (P.24 lines 16- P.25 lines 4).

Query 6: The discussion is poor. The authors should write a proper discussion, including technical aspects (e.g. would repeated application be better, was the timing corret etc.), and aspects of future approaches (do the authors think that their approach does not work at all, why did they not achieve their goals, what improvements can be made or is the concept by itself flawed? Can "better" cell products be achieved? Where do the cells home to?)

Reply:

Thank you for this valuable feedback. In the revised Discussion section, we have addressed several technical aspects that could potentially improve the outcomes of NK cell therapy. We propose several steps to address this: stratifying patients by immunological risk, focusing on low-risk recipients without predisposing factors, such as FcγRIIIA F-carrier status, and increasing infection monitoring for genetically susceptible patients. Adjusting prophylactic treatments, such as broader-spectrum antimicrobials, may be necessary for high-risk individuals. We also plan to conduct a new clinical trial in a low-risk population to evaluate NK cell therapy, while exploring the genetics and functionality of donor NK cells, their interaction with the recipient’s immune system, and strategies to improve NK cell homing and persistence in the liver to enhance infection prevention. These sentences were added in the Discussion section (P.23 lines 7-14). 

Query 7: A lot of abbreviations are not explained in the text; it is unclear what "Sol-Medrol" is and so on. The english of the section "immunological assessments" is not very clear; I do not understand all sentences.

Reply:

We apologize for the lack of clarity in this section. We have now revised the text to clearly define all abbreviations when they first appear in the manuscript. Additionally, the "Immunological Assessments" section has been rewritten to improve its readability and clarity. We have explained the immunological evaluation methods in more detail, including the use of flow cytometry, CFSE-MLR assays, and other techniques used to monitor NK cell activity and donor-specific antibody responses. These revisions should make the section easier to understand.

Thank you for your consideration. I look forward to hearing from you.

Sincerely,

Masahiro Ohira

Department of Gastroenterological and Transplant Surgery, Graduate School of Biomedical and Health Sciences, Hiroshima University, 1-2-3 Kasumi, Minami-ku, Hiroshima city, Hiroshima, Japan 734-8551

Tel: 81-82-257-5222 

Fax: 81-82-257-5224

E-mail: mohira@hiroshima-u.ac.jp

---

## [Decision Letter · Decision Letter 1]

1 Nov 2024

PONE-D-24-28239R1A phase I/II study of adoptive immunotherapy using donor liver graft-derived natural killer cells to prevent severe infection after liver transplantationPLOS ONE

Dear Dr. Ohira,

Thank you for submitting your manuscript to PLOS ONE. After careful consideration, we feel that it has merit but does not fully meet PLOS ONE’s publication criteria as it currently stands. Therefore, we invite you to submit a revised version of the manuscript that addresses the points raised during the review process. As you can see, the reviewer appreciated your work and only minor changes are needed at this stage. Please submit your revised manuscript by Dec 16 2024 11:59PM. If you will need more time than this to complete your revisions, please reply to this message or contact the journal office at plosone@plos.org. Please include the following items when submitting your revised manuscript:A rebuttal letter that responds to each point raised by the academic editor and reviewer(s). You should upload this letter as a separate file labeled 'Response to Reviewers'.A marked-up copy of your manuscript that highlights changes made to the original version. You should upload this as a separate file labeled 'Revised Manuscript with Track Changes'.An unmarked version of your revised paper without tracked changes. You should upload this as a separate file labeled 'Manuscript'.If applicable, we recommend that you deposit your laboratory protocols in protocols.io to enhance the reproducibility of your results. Protocols.io assigns your protocol its own identifier (DOI) so that it can be cited independently in the future. For instructions see: https://journals.plos.org/plosone/s/submission-guidelines#loc-laboratory-protocols. Additionally, PLOS ONE offers an option for publishing peer-reviewed Lab Protocol articles, which describe protocols hosted on protocols.io. Read more information on sharing protocols at https://plos.org/protocols?utm_medium=editorial-email&utm_source=authorletters&utm_campaign=protocols.

We look forward to receiving your revised manuscript.

Kind regards,

Pavel Strnad

Academic Editor

PLOS ONE

Journal Requirements:

Reviewers' comments:

Reviewer's Responses to Questions

**Comments to the Author**

1. If the authors have adequately addressed your comments raised in a previous round of review and you feel that this manuscript is now acceptable for publication, you may indicate that here to bypass the “Comments to the Author” section, enter your conflict of interest statement in the “Confidential to Editor” section, and submit your "Accept" recommendation.

Reviewer #1: All comments have been addressed

Reviewer #2: All comments have been addressed

Reviewer #3: (No Response)

2. Is the manuscript technically sound, and do the data support the conclusions?

Reviewer #1: Yes

Reviewer #2: Yes

Reviewer #3: (No Response)

3. Has the statistical analysis been performed appropriately and rigorously? 

Reviewer #1: Yes

Reviewer #2: Yes

Reviewer #3: (No Response)

4. Have the authors made all data underlying the findings in their manuscript fully available?

Reviewer #1: Yes

Reviewer #2: Yes

Reviewer #3: (No Response)

5. Is the manuscript presented in an intelligible fashion and written in standard English?

Reviewer #1: Yes

Reviewer #2: Yes

Reviewer #3: (No Response)

6. Review Comments to the Author

Reviewer #1: Thank you for addressing all comments made by the reviewers.

In my opinion, the manuscript in its the current version is suitable for publication.

Reviewer #2: I still would suggest changing the title of the manuscript as the cell product is not limited to NK cells

Reviewer #3: This manuscript is a study investigating adoptive immunotherapy using donor liver graft-derived natural killer cells to prevent severe infection after liver transplantation using a phase I/II clinical trial. The following are my comments.

In Statistical analysis, the description of this sentence “NK cell activity at each point was compared using the Mann-Whitney U test” is not clear. Does it mean that to compare NK cell activity between time points? If it is, paired-test or Wilcoxon signed-rank test may be considered.

For Figure 2B, please indicate what statistical test was used to compare the proportions of NK cells before and after activation.

7. PLOS authors have the option to publish the peer review history of their article (what does this mean?). If published, this will include your full peer review and any attached files.

Reviewer #1: No

Reviewer #2: No

Reviewer #3: No

---

## [Author Response · Author response to Decision Letter 1]

1 Nov 2024

Dear Professor, Strnad:

Thank you for providing the reviewers’ insightful comments on our manuscript, titled “A phase I/II study of adoptive immunotherapy using donor liver graft-derived natural killer cells to prevent severe infection after liver transplantation.” We appreciate the opportunity to resubmit our revised manuscript for publication in the PLOS ONE. Below is a detailed point-by-point response to the reviewers' comments. All modifications are marked in red in the revised manuscript.

Response to Reviewer #2

Query 1: I still would suggest changing the title of the manuscript as the cell product is not limited to NK cells.

Reply:

The title has been changed to “A phase I/II study of adoptive immunotherapy using donor liver graft-derived NK cell-enriched immune cells to prevent severe infection after liver transplantation.”

Response to Reviewer #3

Query 1: In Statistical analysis, the description of this sentence “NK cell activity at each point was compared using the Mann-Whitney U test” is not clear. Does it mean that to compare NK cell activity between time points? If it is, paired-test or Wilcoxon signed-rank test may be considered.

Reply:

The sentence "NK cell activity at each point was compared using the Mann-Whitney U test, " which refers to comparisons of NK cell activity levels between time points for unpaired samples. However, as this study involved a repeated-measures design in which the same individuals were assessed at multiple time points, we recognize that a paired analysis would be more appropriate. We revised our statistical approach to use the Wilcoxon signed-rank test, which is more suitable for paired data. We updated this in the Methods section and adjusted the analysis accordingly. These details have been added to the Methods section (page 11, lines 16 – 12, lines 2) and the Results section (page 21 lines 11-12).

Query 2: For Figure 2B, please indicate what statistical test was used to compare the proportions of NK cells before and after activation.

Reply:

Thank you for pointing out the need to specify the statistical test used to compare the NK cell proportions before and after activation. As shown in Figure 2B, we used a paired t-test to assess the statistical significance of differences in NK cell proportions. We have clarified this in the figure legend and added this information to the Methods section (page 11 lines 16–12 lines 2, page 16 lines 11-12).

Thank you for your consideration of our manuscript. We look forward to hearing from you.

Sincerely,

Hideki Ohdan

Department of Gastroenterological and Transplant Surgery, Graduate School of Biomedical and Health Sciences, Hiroshima University, 1-2-3 Kasumi, Minami-ku, Hiroshima city, Hiroshima, Japan 734-8551

Tel: 81-82-257-5220

Fax: 81-82-257-5224

E-mail: hohdan@hiroshima-u.ac.jp

---

## [Decision Letter · Decision Letter 2]

14 Nov 2024

A phase I/II study of adoptive immunotherapy using donor liver graft-derived NK cell-enriched immune cells to prevent severe infection after liver transplantation

PONE-D-24-28239R2

Dear Dr. Ohira,

We’re pleased to inform you that your manuscript has been judged scientifically suitable for publication and will be formally accepted for publication once it meets all outstanding technical requirements.

Kind regards,

Pavel Strnad

Academic Editor

PLOS ONE

Additional Editor Comments (optional):

Reviewers' comments:

Reviewer's Responses to Questions

**Comments to the Author**

1. If the authors have adequately addressed your comments raised in a previous round of review and you feel that this manuscript is now acceptable for publication, you may indicate that here to bypass the “Comments to the Author” section, enter your conflict of interest statement in the “Confidential to Editor” section, and submit your "Accept" recommendation.

Reviewer #3: All comments have been addressed

2. Is the manuscript technically sound, and do the data support the conclusions?

Reviewer #3: (No Response)

3. Has the statistical analysis been performed appropriately and rigorously? 

Reviewer #3: (No Response)

4. Have the authors made all data underlying the findings in their manuscript fully available?

Reviewer #3: (No Response)

5. Is the manuscript presented in an intelligible fashion and written in standard English?

Reviewer #3: (No Response)

6. Review Comments to the Author

Reviewer #3: (No Response)

7. PLOS authors have the option to publish the peer review history of their article (what does this mean?). If published, this will include your full peer review and any attached files.

Reviewer #3: No

---

## [Editor Report · Acceptance letter]

3 Dec 2024

PONE-D-24-28239R2 

PLOS ONE

Dear Dr. Ohira, 

I'm pleased to inform you that your manuscript has been deemed suitable for publication in PLOS ONE. Congratulations! Your manuscript is now being handed over to our production team.

Kind regards, 

on behalf of

Dr. Pavel Strnad 

Academic Editor

PLOS ONE